# Dynamics of Microbial Communities, Flavor, and Physicochemical Properties during *Ziziphus jujube* Vinegar Fermentation: Correlation between Microorganisms and Metabolites

**DOI:** 10.3390/foods11213334

**Published:** 2022-10-24

**Authors:** Wei Ruan, Junli Liu, Pengliang Li, Wei Zhao, Aixia Zhang, Songyan Liu, Zhixin Wang, Jingke Liu

**Affiliations:** 1Institute of Biotechnology and Food Science, Hebei Academy of Agriculture and Forestry Sciences, 598 Heping West Road, Shijiazhuang 050031, China; 2College of Food and Biology, Hebei University of Science and Technology, 26 Yuxiang Street, Shijiazhuang 050000, China; 3Shijiazhuang Quality Inspection Centre of Animal Products, Feed, and Veterinary Drugs, 3 Yixi Street, Shijiazhuang 050035, China

**Keywords:** *Ziziphus jujube* vinegar, high-throughput sequencing, microorganism, flavor substance, correlation

## Abstract

Jujube pulp separated from *Ziziphus jujube* is often discarded after processing, resulting in a serious waste of resources and environmental pollution. Herein, *Ziziphus jujube* pulp was used as a raw material for vinegar fermentation. To investigate the dynamic distribution of microorganisms and flavor substances in ZJV, correlations between environmental variables (e.g., total acid, reducing sugar, temperature) and flavor substances (organic acids, amino acids, volatile substances) and microorganisms were analyzed. Physicochemical indicators (temperature, total acid, alcohol) were the main factors affecting ZJV fermentation. The middle and later stages of ZJV fermentation were the periods showing the largest accumulation of flavor substances. Organic acids (acetic acid, malic acid, citric acid, lactic acid), amino acids (Asp, Glu, Arg) and volatile substances (ethyl phenylacetate, phenethyl alcohol) were important odor-presenting substances in ZJV. In the bacterial community, the Operational Taxonomic Units (OTUs) with an average relative abundance of more than 10% in at least one fermentation stage were mainly *Acetobacter*, *Lactobacillus* and *Saccharopolyspora*, while it was *Thermomyces* in the fungal community. Pearson correlation coefficients showed that *Penicillium*, *Lactobacillus* and *Acetobacter* were the core microorganisms, implying that these microorganisms contributed to the flavor formation greatly in ZJV fermentation. This study reveals the correlation between physicochemical indexes and flavor substances and microorganisms in ZJV fermentation. The results of the study can provide a theoretical basis for the development of the ZJV industry.

## 1. Introduction

As a condiment, vinegar is one of the oldest fermentation foods in the world and has been used for more than 10,000 years [1]. Studies demonstrate that vinegar has the effects of antioxidants, fatigue recovery and regulating blood pressure and blood sugar [2,3]. Thus, various vinegars have been investigated recently.

Traditionally, vinegars are mostly fermented from grains, and sorghum, rice and foxtail millet are the main raw materials for vinegar fermentation [4,5]. However, these grains mainly contain high levels of starch and lack functional components. In recent years, vinegar made from fruits has received more attention from scientists.

*Ziziphus jujuba* (*Ziziphus jujuba* Mill.) belongs to the Rhamnaceae plant [6], and has been widely planted in Europe and Asia. In China, the jujuba fruit is an important wild economic plant [7], and the jujube pit is used as a type of Chinese herb for curing insomnia. However, the jujube pulp is often discarded. It has been shown that the jujube pulp is rich in nutrients and contains a large amount of carbohydrates, fat, protein, vitamins, phenolics and minerals [8]. Therefore, the jujube pulp can be used as a raw material for vinegar fermentation, which can reduce the waste of *Ziziphus jujuba*.

The flavor (aroma and taste) is an essential index for evaluating vinegar quality, which influences the consumer’s satisfaction. The aroma, mainly originating from volatile compounds, contains acids, aldehydes, alcohols, esters, ketones and phenols [9], while the taste mainly includes organic acids and free amino acids [9]. The flavor compounds are mainly formed during vinegar fermentation. Generally speaking, the solid-state fermentation of vinegar leads to a more intense flavor and better taste than liquid-state fermentation. The reason is that the solid-state fermentation of vinegar has more complex enzyme systems, and these microorganisms play an important role in the formation of flavor compounds [10].

At present, culture-dependent analysis of microorganisms during vinegar fermentation, such as polymerase chain reaction denaturing gradient gel electrophoresis, only covers the easily culturable microorganisms, which is not sufficient to understand the microbial community. High-throughput sequencing technology has developed rapidly due to its high safety and accuracy. It is increasingly used to analyze the abundance and diversity of microorganisms in samples. For example, the correlation between microbial succession and metabolite changes in Shanxi aged vinegar was explored using high-throughput sequencing. The results showed that *Saccharomyces* was positively correlated with alcohols and might contribute to alcohol production. *Acetobacter* and *Komagataeibacter* promoted the conversion of alcohols to acetic acid [11]. *Trichoderma* contributed to the degradation of cellulose and hemicellulose, promoted their conversion to fermentable monosaccharides and transformed them to phenylethyl acetate from β-phenylethanol and acetic acid, whereas *Lactobacillus*, *Acetobacter* and *Rhodobacter* contributed to the organic acids and volatiles in grain vinegar [12]. However, there are fewer studies on the correlation between microorganisms and flavor substances during the fermentation of fruit vinegars.

In this paper, *Ziziphus jujuba* has been used as a raw material for the fermentation of vinegar. The dynamic changes in taste substances (organic acids, amino acids) and aroma components (volatile substances) during *Ziziphus jujuba* vinegar (ZJV) fermentation were investigated. The microbial community composition and succession during ZJV fermentation were also detected using the Illumina MiSeq platform with 16S rRNA and ITS gene sequencing technology. The aim of this study was to show the correlation of microbial communities and flavor metabolites during ZJV fermentation, which can provide a theoretical basis for the quality improvement of ZJV.

## 2. Materials and Methods

### 2.1. Ziziphus jujube Vinegar Fermentation and Sample Collection

The *Ziziphus jujube* pulp (total sugar, 40.6 g/100 g) was provided by Shijiazhuang Yiling Pharmaceutical Co., Ltd. (Shijiazhuang, China), and the vinegar fermentation was performed at Hebei Su Ning Xiang Food Co., Ltd. (Shijiazhuang, China). Briefly, 25 kg *Ziziphus jujube* pulp was mixed with 50 L water in a plastic film-covered fermentation tank at a fluctuating room temperature. The fermentation was divided into two stages, including 7-day alcohol fermentation and 14-day acetic acid fermentation. Five kilograms of Daqu (a type of microbe complex) was added for 7-day alcohol fermentation. Afterwards, 25 g acetic acid bacteria and 10 kg accessory materials (brans of wheal) were added and the tank was covered with a breathable straw mat. The mixture was stirred once every day to ensure sufficient oxygen in the system. The acetic acid fermentation took 14 days. Samples were collected in four positions approximately 10–20 cm below the fermenter surface and then thoroughly mixed for a fermentation period of 1, 7, 12, 17 and 22, named ZJV1, ZJV2, ZJV3, ZJV4 and ZJV5, respectively. Triplicates were prepared for every sample collection and stored at −80 °C.

### 2.2. DNA Extraction and PCR Amplification

The total DNA of ZJV samples was extracted using the Fast DNA^®^ Spin Kit for Soil DNA Extraction Kit (MP Biomedicals, Santa Ana, CA, USA) and the extracted qualified DNA was amplified by PCR. The quality of extraction was examined by 1% agarose gel electrophoresis. The hypervariable region V3–V4 of bacterial 16S rRNA was amplified with primers 338F (5’-ACTCCTACGGGAGGCAGCAG-3’) and 806R (5’-GGACTACHVGGGTWTCTAAT-3’). The ITS1-ITS2 region of fungal ITS was amplified with primers ITS1F (5’-CTTGGTCATTTAGAGGAAGTAA-3’) and ITS2R (5’-GCTGCGTTCTTCATCGATGC-3’).

### 2.3. High-Throughput Sequencing and Statistical Analysis of Biological Information

The paired-end (PE) sequence data were obtained with an Illumina MiSeq PE300 platform/NovaSeq PE250 platform (Illumina, San Diego, CA, USA). According to the overlap relationship between PE reads, the paired reads were merged into a sequence and filtered for quality. The effective sequences were obtained by distinguishing samples according to barcode and primer sequences, and the sequence orientation was corrected to obtain optimized sequences. Bioinformatic statistical analysis was performed on OTUs at a 97% similarity level.

### 2.4. Physicochemical Analysis

The temperature was measured using a sterilized thermometer by inserting it at 10–20 cm into the sample in the tank. pH was measured using a pH meter (Ultra Basic Benchtop, Denver Instrument, Bohemia, NY, USA). The Brix degree was detected using a refractometer (WYT-1, Shanghai Precision Instrument Company, Shanghai, China). Alcohol was determined using a vinometer after distillation. The content of reducing sugars in vinegar was determined using the 3,5-dinitrosalicylic acid method (DNS method) [13]. The determination of total acid content was performed using the method of Wu et al. Total acid of the phenolphthalein endpoint was titrated with 0.1 mol/L NaOH. [14]. All measurements were performed in triplicate.

### 2.5. Determination of Organic Acids (OAs) and Free Amino Acids (FAAs)

Determination of organic acids was performed according to a previous method, with slight modification [15]. Briefly, a high-performance liquid chromatography (2695 HPLC, Waters, Milford, MA, USA) system with an AQ-C18 (4.6 mm × 150 mm × 5 μm) column was used for organic acid analysis. Organic acid standards (acetic acid, lactic acid, citric acid, malic acid, tartaric acid, α-ketoglutaric acid) were used for the quantification of organic acids in ZJV samples. First, we used 20 mmol/L NaH_2_PO_4_ with a flow rate of 0.7 mL/min as the mobile phase. The injection volume was set to 10 μL. The UV detector wavelength was 210 nm and the column temperature was 30 °C. The free fatty acids were detected by an automatic amino acid analyzer according to an in-line method. The vinegar was filtered with a 0.22 µm membrane for analysis [16].

### 2.6. Volatile Analysis

The volatiles of the vinegar were detected using HS-SPME-GC-MS. Briefly, 8 mL of vinegar sample and 2 g of NaCl were mixed in a 20 mL screwed headspace vial. The absorption of the volatiles took place for 40 min at 60 °C, using a 75 μm CAR/PDMS fiber. A DB-5MS capillary column (30 m × 0.25 mm × 0.25 μm, Agilent, St. Clara, CA, USA) was used for the separation of the volatiles. The initial temperature was 40 °C for 2 min, and the sample was then heated to 240 °C at 5 °C/min and held for 5 min. The injection port temperature was 250 °C with splitless mode, and the carrier gas was helium (He) at a flow rate of 1.0 mL/min. Electron ionization (EI) was used with energy of 70 eV. The ion source temperature and ion interface temperature were set to 230 °C and 250 °C, respectively. The mass scan range was 33–450 m/z.

### 2.7. Statistical Analysis

Rarefaction curves of microbial communities, taxonomic diversity of microbial communities and Spearman correlation analysis of microbiota and physicochemical indicators were plotted using Origin 2022 (OriginLab, Northampton, MA, USA). Principal component analysis (PCA) was performed on volatile flavor data, and the upsetR package in R v4.0.2 (R Foundation for Statistical Computing, Vienna, Austria) was used to show all possible interactions. Two-way orthogonal partial least squares (O2PLS) was developed using SIMCA 14.1 (UMETRICS, Umeå, Sweden), and the correlation coefficients were obtained via the relationships between the 50 flavor substances and microorganisms (29 bacteria and 24 fungi, relative abundance >1% in at least one stage during fermentation). The network diagram between flavor substances and microbial communities was visualized using Cytoscape 3.9.1 (Cytoscape Team, Seattle, WA, USA) (*p* < 0.05 and |r| > 0.85).

## 3. Results and Discussion

### 3.1. Changes in Physicochemical Properties during ZJV Fermentation

The physicochemical properties, including temperature, pH, total acid, reducing sugars, alcoholic content and soluble solids, of the samples are usually used to detect the fermentation of ZJV [17]. As shown in Figure 1, the total acid content increased sharply from 1.13 ± 0.05 g/100 mL to 5.81 ± 0.11 g/100 mL. The alcohol was increased to 5.84 ± 0.07% after alcohol fermentation (ZJV 2), and then it dramatically decreased during the entirety of acetic acid fermentation (ZJV 2–5) because the alcohol was transformed into acetic acid by *Acetobacter*. The pH value was maintained at 3.3–3.5 for the whole fermentation stage. The initial temperature of fermentation was around 22 °C and reached 44 °C during the acetic acid fermentation stage, and it then decreased slightly. The content of reducing sugars began to grow at the ZJV3 stage and reached 107.17 ± 1.90 g/L. This phenomenon was not in accordance with the result obtained for cereal vinegar, in which the reducing sugars consistently decreased during vinegar fermentation. The reason might be the fact that the jujube contained a high level of polysaccharides, which were degraded by the microorganisms existing at the fermentation stage [18]. The high level of reducing sugars would be helpful for the good flavor formation of jujube vinegar.

### 3.2. ZJV Taste Substance Analysis

Amino acids and organic acids can soften the taste of ZJV and are important indicators of the quality thereof for consumers. Amino acids are important nutrients, as well as important components for taste. In general, amino acids are classified into fresh, sweet, bitter and astringent flavors [19]. Among the 18 free amino acids detected in ZJV (Appendix A), there were five sweet ones, Thr, Ser, Gly, Ala and Pro, among which Pro had the highest content (92.29 mg/100 mL). Fresh amino acids are Asp and Glu [20]. Bitter amino acids are Val, Ile, Leu, Phe, Lys, His and Arg; among these, Phe, Val, Lys and Leu had the highest content, of 26.98, 26.55, 22.42 and 48.69 mg/100 mL, respectively. Tyr, an astringent amino acid, was also present, but had a value of only 8.91 mg/100 mL. We identified three amino acids, Asp, Glu and Arg, which contributed to the taste of ZJV (taste activity value, TAV > 1). The content of the remaining amino acids in ZJV was low, and they mainly played a role in softening its flavor. During the pre–mid-fermentation period, because the proteins in the raw materials were decomposed by proteases, and due to the addition of auxiliary materials, these substances decomposed and were metabolized under the action of microorganisms; thus, amino acids accumulated gradually, with the levels remaining largely constant during the late fermentation period. In addition, γ-aminobutyric acid (GABA), a functional amino acid that lowers blood pressure and improves brain function [21], was detected in ZJV at 8.31 mg/100 mL. The accumulation of free amino acids not only improved the flavor of ZJV, but also provided the precursors necessary for the Maillard reaction in ZJV in the subsequent aging stage; this promoted the formation of flavor and functional substances in ZJV, thereby providing a more balanced flavor. Most of the organic acids in vinegar are converted from proteins, starches and fats in the raw materials through the action of microbes during fermentation [22]. In this study, seven organic acids in ZJV were detected by HPLC (Table 1), and the results showed that the total organic acid content increased in the early–middle stage, and then remained steady. The TAVs of organic acids were calculated based on the threshold values (Appendix A), and the results showed that all the values were >1, indicating that these organic acids contribute to the flavor of ZJV [19]. In particular, acetic acid is an important taste-presenting organic acid with a pungent taste and short aftertaste [23], and it had a TAV > 100 and a quantity of 2081.27 ± 24.65 mg/100 mL in ZJV; it and was the main source of the taste of ZJV. Among the organic acids reported in most of the grain vinegars, acetic acid and lactic acid were the main ones [24]. It is important to note, however, that citric acid and malic acid were higher in quantity than lactic acid in ZJV, and their content levels were 320.34 ± 18.66, 265.00 ± 10.15 and 149.25 ± 15.24 mg/100 mL, respectively. Their presence reduced the influence of acetic acid and increased the crispness of ZJV.

### 3.3. ZJV Aroma Composition Analysis

The raw materials of vinegar and various microbial metabolites produced during the fermentation process, as well as substances transformed and generated during the subsequent aging stage, were the main sources of volatile components in the vinegar. A total of 25 volatile flavor compounds were detected by HS-SPME-GC-MS during the fermentation of ZJV (Figure 2a), including nine esters, five acids, two alcohols, five aldehydes, one phenol and three heterocyclic substances. Principal component analysis of the samples (Figure 2c) showed that the contribution of the first principal component was 57.3%, while that of the second principal component was 23.8% (combined contribution of 81.1%). The flavor substances of the five fermentation stages appeared to be well separated, indicating differences in volatile flavor substances. Combined with Figure 2b, the results showed that most flavor substances were formed after the third stage of fermentation, and the types of flavor substances gradually increased as fermentation proceeded. Ester substances are generated by the esterification of alcohols and acids, mostly via biochemical reactions of yeast and *Aspergillus*. The acids are activated by the acyl coenzyme A and then synthesized with ethanol to the corresponding ethyl esters, i.e., ethyl benzoate, ethyl phenylacetate, phenethyl acetate and ethyl palmitate, which confer fruity flavors to vinegars. Acetic acid is the main volatile acid produced by *Acetobacter* from ethanol under sufficient oxygen. It was found in the middle and later stages of fermentation and promoted the accumulation of a sour taste in ZJV. Phenethyl alcohol is synthesized by *Saccharomyces* through the shikimic acid pathway and the Ehrlich pathway [25]; as the main volatile alcohol with floral and rose aromas in ZJV [26], its content gradually increased during fermentation and softened the taste of ZJV. Moreover, 2,3,5,6-tetramethylpyrazine was also found in ZJV, which not only provides nutty, coffee and chocolate flavors, but also has antioxidant, anti-inflammatory and anti-apoptotic properties that are beneficial for the treatment of cardiovascular and neurological diseases [27]. These volatile flavor substances are present in vinegar in very small amounts, but they can impart unique aromatic notes to vinegar and different aroma characteristics in different combinations and proportions.

### 3.4. Alpha Diversity Analysis

The abundance and diversity of microbial species in the 15 ZJV samples were evaluated by using the ACE index, Shannon index and Simpson index at 97% similarity of OTUs. The results showed that both the ACE index and Shannon index reached the maximum, while the Simpson index was at the minimum at the ZJV 2 stage (Table 2), which indicated that both the OTUs and community diversity reached the highest level. The reason might be associated with the addition of the wheal brans during the fermentation of ZJV. The coverage of all samples was greater than 99%, indicating that the identification of the microorganisms was highly possible. The sample was sufficient for sequencing according to the rarefaction curves, which reached flatness. Figure 3 shows that the samples of the bacterial and fungi could meet the requirement for subsequent bioinformatic analysis.

### 3.5. Analysis of Microbial Composition

The bar chart of the community composition reflects the dominant species and their relative abundance in different samples. Firmicutes (48.50%) and Proteobacteria (38.99%) were the dominant bacteria during the fermentation of ZJV (Figure 4a), and Firmicutes played an important role in the early stage of fermentation. Proteobacteria gradually increased at the origin stage of acetic acid fermentation (ZJV2), while Firmicutes decreased. The reason might be the antagonistic effect of the two bacteria. At the final fermentation stage (ZJV5), Proteobacteria decreased while Firmicutes increased. The high level of acid might inhibit the growth of Proteobacteria [28].

In comparison, Ascomycota (94.56%) and Basidiomycota (5.04%) were the dominant phyla in the fungal community throughout fermentation (Figure 4c). This phenomenon is consistent with the results of cereal vinegar fermentation [22,29], indicating that Ascomycota and Basidiomycota might play important roles in vinegar fermentation.

Moreover, the top three bacteria at the genus level throughout ZJV fermentation mainly included *Acetobacter* (31.20%), *Virgibacillus* (22.69%) and *Lactobacillus* (14.60%) (Figure 4b). *Acetobacter* was the main bacterium to produce acetic acid, and it rapidly increased to 66.21% during ZJV fermentation. The high level of acetic acid produced a low-pH environment in the fermentation system, and resulted in a decrease in some genera [28]. The relative abundance of *Lactobacillus* reached 35.07%, and it was also decreased with the change in the fermentation environment [30]. In addition, *Virgibacillus* was unexpectedly found throughout the fermentation process of ZJV. *Virgibacillus* was a microorganism presenting in Daqu, and it had acid and salt tolerance and degraded the protein to flavor precursors (peptides, amino acids) by producing protease [31]. Studies also show that it has high potential to improve the volatile components, glutamate content and overall acceptability of the product when applied to fermented food [32].

*Aspergillus* (42.97%) and *Thermomyces* (8.92%) were the main fungi present throughout the fermentation of ZJV (Figure 4d). *Aspergillus* and *Thermomyces* can produce a large number of highly active proteases and glycosylases, which can degrade the protein and starch for nitrogen sources and carbon sources [33].

### 3.6. Correlations between Microbiota and Flavor Substances and Physicochemical Indicators during ZJV Fermentation

Microorganisms play an important role in the production of physicochemical indicators and flavor substances, but the correlations between them are not clear. Therefore, Spearman correlation analysis was conducted between microorganisms (seven bacteria and six fungi, average relative abundance >10% in at least one stage) and physicochemical indicators (Figure 1b). Dominant OTUs during fermentation with taxonomic annotation was shown in Table 3. OTU314 (*Acetobacter*) and OTU325 (*Lactobacillus*) showed a significant positive correlation with temperature (*p* < 0.05), which indicated that an appropriate fermentation temperature promotes the growth and reproduction of microorganisms. OTU314 showed a significant negative correlation with pH (*p* < 0.05), and OTU5 (*Saccharopolyspora*), ZOTU12 (*Thermomyces*) and ZOTU167 (*Xeromyces*) all inhibited the production of soluble solids and total acid. ZOTU530 (*Aspergillus*) showed a highly significant positive correlation (*p* < 0.01) with reducing sugars, and it can produce a variety of carbohydrate enzymes, especially starch-degrading enzymes that suppress sugar production [18]. ZOTU333 (*Cladosporium*) and ZOTU276 (*Alternaria*) were significantly and positively correlated with alcohol (*p* < 0.05 and *p* < 0.01, respectively). *Cladosporium* degrades acetaldehyde, 1-glutaraldehyde, etc., in wines, accelerates aging, promotes ester production and contributes positively to flavor [34].

The correlation between microorganisms and flavor substances during ZJV fermentation was analyzed via O2PLS, and the values for the three indicators used to evaluate the fit of the O2PLS model (R2X, R2Y and Q2Y) were 1, 0.997 and 0.996, respectively; these results indicated an almost perfect fit for analysis and prediction [35]. The *VIP* (pred) vector (variable importance in projection) was applied to measure the intensity and explanatory power of the microorganisms in the formation of flavor metabolites (Figure 5b). There were 10 bacterial and 12 fungal OTUs with *VIP* values >1, indicating that these microorganisms are the major strains affecting flavor metabolites. Among these OTUs, five had an average relative abundance >10%: OTU5, OTU314, ZOTU12, ZOTU167 and ZOTU530. Furthermore, 11 and 12 flavor compounds were associated with OTU314 and OTU5, respectively, and 17, 9 and 8 with ZOTU12, ZOTU167 and ZOTU530, respectively (Figure 5a), indicating the important effects of fungi on flavor metabolites. As shown in Figure 5c, OTU314, OTU335 (*Lactobacillus*), ZOTU204 (*Cladosporium*), ZOTU357 (*Fusarium*), ZOTU450 (*Penicillium*) and ZOTU450 (*Lactobacillus*) had a significant role in flavor metabolism. ZOTU450 and ZOTU530 were positively correlated with the formation of most organic acids and free amino acids. Among them, OTU335 showed a significant positive correlation (|r| > 0.9) with malic acid, citric acid and other organic acids, mainly through participation in the tricarboxylic acid cycle (TCA cycle, wherein citric acid is first generated from acetyl CoA and oxaloacetate, and malic acid is synthesized after four dehydrogenation processes [36]). This pathway promotes the production of malic acid, citric acid, etc., which makes ZJV milder and provides a refreshing taste. In addition, ZOTU204, ZOTU450 and ZOTU530 contribute to the production of citric acid, tartaric acid, etc., mainly by metabolizing glucose [37]. When the acidity reaches a certain level, aconitate hydratase becomes inactive, and the TCA cycle is blocked; this allows for the substantial accumulation of citric acid [38].

OTU314 and ZOTU450 were highly correlated (|r| > 0.8) with eight and seven amino acids, respectively, and were the major producers of amino acids. Among the amino acids, Asp, Glu and Arg made the most important contributions to the flavor of ZJV, and Asp was strongly correlated (|r| > 0.9) with OTU287, ZOTU450 and ZOTU450, which all promoted Asp production. Glu, which provides a fresh flavor, was closely related to five OTUs, where OTU335 and ZOTU450 were highly correlated with changes in Glu (|r| > 0.8). During ZJV fermentation, the glutaminase of OTU335 promotes the accumulation of Glu, which in turn contributes to the umami of fermented foods [36]. Moreover, α-ketoglutaric acid, an intermediate product of the TCA cycle, also acts as a synthetic precursor of Glu to promote glutamate production [39]. Arg, which provides a bitter taste, was highly correlated with four bacterial OTUs (|r| > 0.9) in this study, and exhibited positive correlations with OTU324, OTU325 and OTU48 (all of which belong to *Lactobacillus*); the reason for this may be the ability of *Lactobacillus* to hydrolyze peptides with specific aminopeptidases [40]. The bioactive substance GABA, which is present in ZJV, has physiological effects, i.e., antihypertensive, antioxidant and sedative effects [41], and changes in GABA during fermentation were highly correlated with four OTUs in this study (all of which were bacteria). This result is consistent with a previous study [42]. There was a positive correlation (|r| > 0.9) with OTU314, indicating a role of *Acetobacter* in promoting functional amino acid production.

The direct contribution of free amino acids to flavor is limited, but they can serve as precursors for the formation of other flavor substances [43]. Amino acids can be decarboxylated to form amines and CO_2_, and deaminated to produce ammonia and α-keto acids (which are further converted to aldehydes, alcohols, acids, sulfur-containing compounds, indoles, phenols, etc. [44]). For example, Trp generates indole pyruvate by the action of transaminase, which is further catabolized to benzaldehyde [40]. Regarding volatile flavor substances, fungi play a more important role than bacteria. ZOTU450, ZOTU530, ZOTU204 and OTU335 are essential for the production of volatile flavor substances, and are strongly associated with 10 VFs. In this study, OTU335 was highly correlated with phenylethyl alcohol and ethyl phenylacetate, which are derived from Phe [40]. ZOTU530 was significantly and positively correlated with the formation of furfural, which is bacteriostatic [45] and can be produced by dehydration and the cyclization of pentose [46], as well as through the formation of several key organic acids (acetic acid, hexanoic acid, decanoic acid, dodecanoic acid) that react with alcohols to produce ester-like aroma components. In addition, ZOTU450, ZOTU165, OTU335 and ZOTU204 correlated well with changes in 2,3,5,6-tetramethylpyrazine, a functional bioactive agent that not only contributes to the flavor of vinegar but also has beneficial effects on vasodilation and lowers lipid levels and antioxidant activity [47].

## 4. Conclusions

To obtain a comprehensive picture of the factors affecting the fermentation flavor of ZJV, we studied the physicochemical indexes and flavor substance formation pattern during the whole process of ZJV fermentation, and investigated their correlations and microbiota at different fermentation stages. Total acid, alcohol and temperature were important environmental factors affecting the structure of the microbial community during fermentation. The middle and later stages of fermentation were the main stages of flavor substance accumulation, and the content of some flavor substances continued to rise, indicating the positive effect of this stage. Both bacteria and fungi played a non-negligible role in promoting flavor formation, especially *Penicillium*, *Lactobacillus* and *Acetobacter*. Previous studies show that bacteria play an important role in vinegar fermentation. However, in future studies, the roles of fungi during the vinegar fermentation should be further investigated. This study is the first report to explore the relationships between the microorganisms, physicochemical indicators and flavor compounds of ZJV. The results provide a scientific basis for the comprehensive utilization of *Ziziphus jujube* pulp resources and lay a foundation for the establishment of a controlled fermentation system and the improvement of ZJV quality.

## Figures and Tables

**Figure 1 foods-11-03334-f001:**
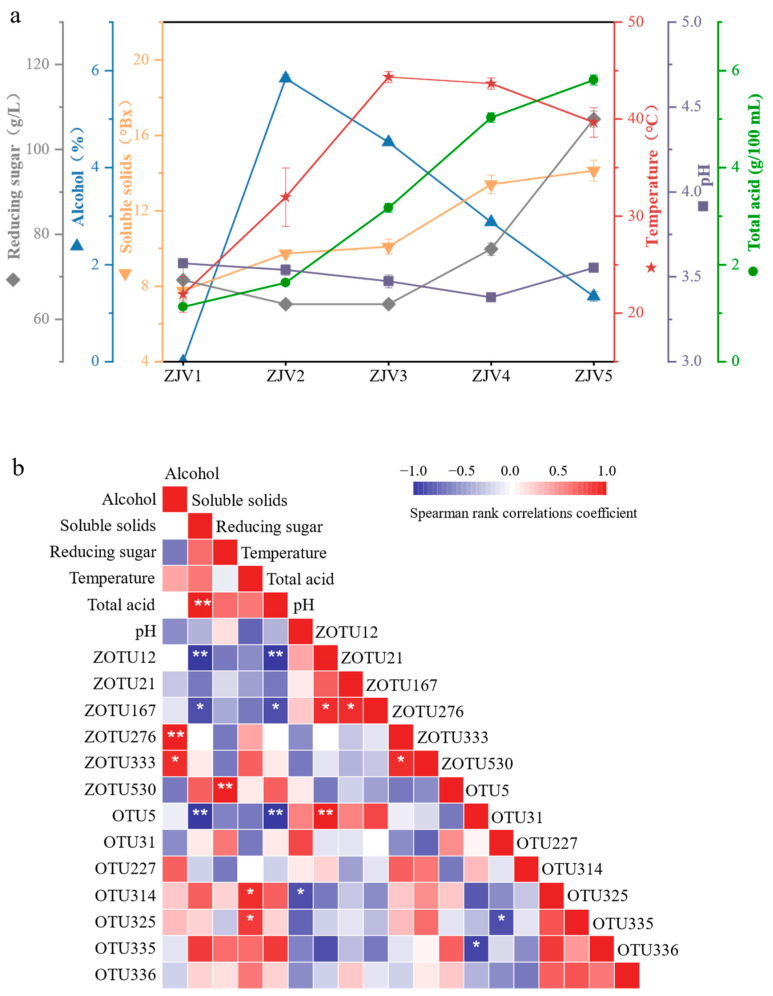
(**a**) Dynamic changes in physicochemical parameters during fermentation of ZJV. Temperature, pH, reducing sugar content, alcohol content, soluble solids and total acidity. (**b**) Heat map of physicochemical indicators and microbial correlation. * and ** mean significance at the 0.05 and 0.01 probability levels respectively.

**Figure 2 foods-11-03334-f002:**
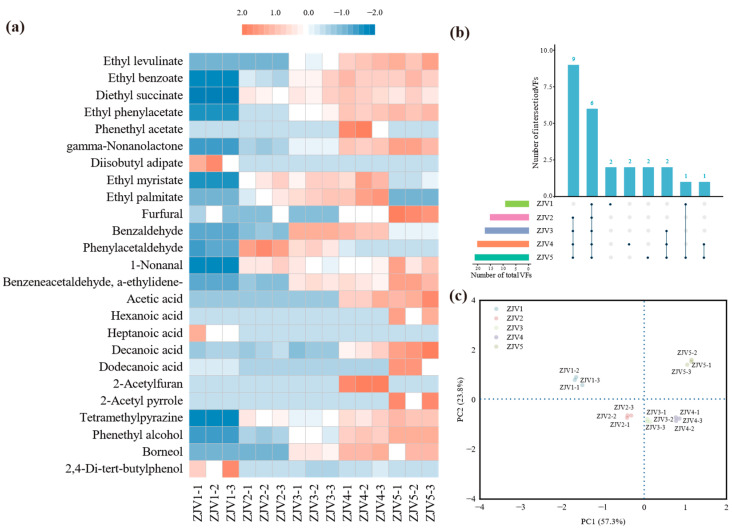
(**a**) Heat map of flavoring substances and microbial correlation. (**b**) UpsetR diagram representing shared and unique flavor substances during fermentation. (**c**) Principal component analysis of volatiles in five fermentation stages.

**Figure 3 foods-11-03334-f003:**
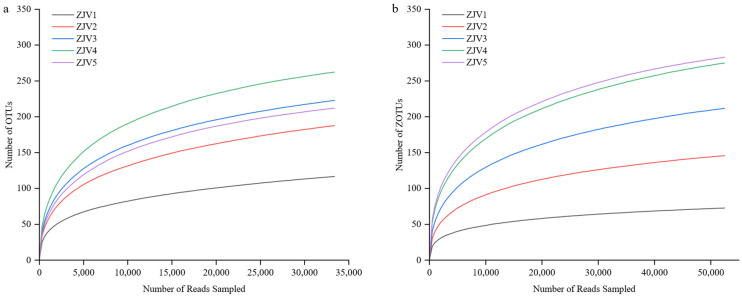
Rarefaction curves for bacteria (**a**) and fungi (**b**).

**Figure 4 foods-11-03334-f004:**
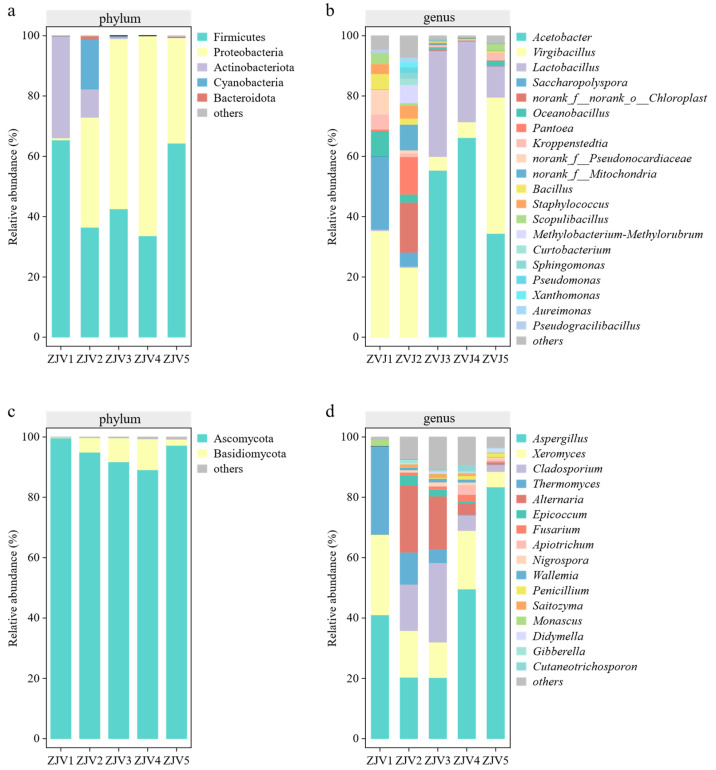
Species diversity and abundance of ZJV at the level of phylum ((**a**) bacteria; (**c**) fungi) and genus ((**b**) bacteria; (**d**) fungi).

**Figure 5 foods-11-03334-f005:**
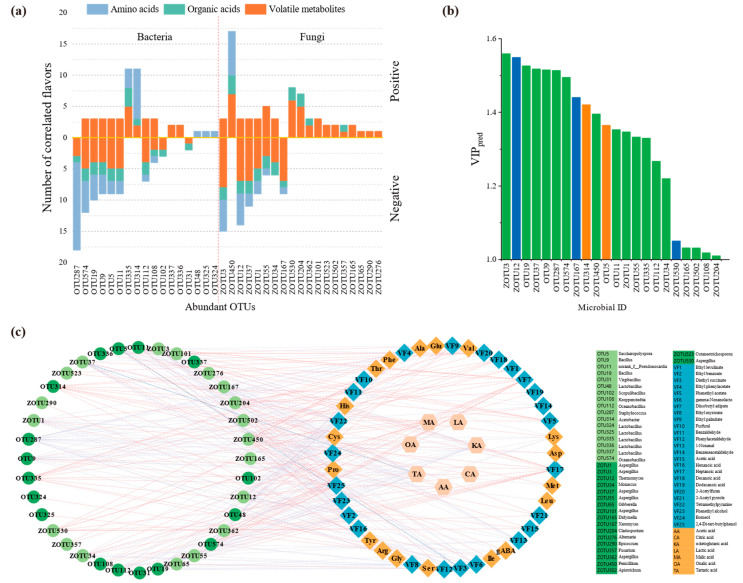
Correlation analysis between microbiota and flavor during fermentation. (**a**) Summary of significantly related flavors. (**b**) OTUs with *VIP* (pred) > 1 by O2PLS analysis. Blue and orange refer to fungal and bacterial OTUs with average relative abundance > 10% in at least one stage, respectively. (**c**) Correlation networks between microbiota and flavor during fermentation. Red and blue lines represent positive and negative correlations, respectively.

**Table 1 foods-11-03334-t001:** Changes in the organic acid fractions of ZJV during fermentation.

Sample	ZJV1	ZJV2	ZJV3	ZJV4	ZJV5	ZJV
Acetic acid	-	36.70 ± 1.84	136.69 ± 19.83	813.02 ± 16.59	1268.86 ± 24.57	2081.27 ± 24.65
Lactic acid	202.41 ± 7.84	631.55 ± 4.56	657.12 ± 16.74	861.19 ± 5.90	82.51 ± 5.11	149.25 ± 15.24
Oxalic acid	30.7 ± 0.32	26.69 ± 2.60	38.27 ± 1.80	35.34 ± 2.89	33.48 ± 0.32	34.70 ± 1.35
Malic acid	33.68 ± 1.34	30.84 ± 6.37	28.00 ± 3.21	115.65 ± 14.65	157.68 ± 7.76	265.00 ± 10.15
Tartaric acid	12.90 ± 0.53	31.85 ± 1.38	36.19 ± 1.92	37.68 ± 0.77	38.87 ± 1.74	56.74 ± 4.72
Citric acid	-	-	-	187.53 ± 34.46	179.89 ± 8.61	320.34 ± 18.66
α-ketoglutaric acid	-	2.59 ± 0.09	4.09 ± 0.63	3.39 ± 0.19	2.93 ± 0.04	5.58 ± 0.52

The results are expressed as mean ± standard deviation (mg/100 mL).

**Table 2 foods-11-03334-t002:** Alpha diversity of bacterial and fungal communities in samples.

Sample ID	Bacteria					Fungi				
	Number of OTUs	Shannon	Simpson	Ace	Coverage (%)	Number of OTUs	Shannon	Simpson	Ace	Coverage (%)
ZJV1_1	81	1.60	0.32	101.95	99.94	62	1.70	0.26	73.92	99.98
ZJV1_2	147	2.40	0.17	232.46	99.87	74	2.04	0.17	83.18	99.98
ZJV1_3	122	2.42	0.14	204.75	99.89	82	2.09	0.16	104.92	99.96
ZJV2_1	294	2.96	0.11	363.25	99.78	281	2.72	0.13	353.73	99.86
ZJV2_2	252	3.21	0.09	282.91	99.87	272	2.73	0.13	333.56	99.88
ZJV2_3	241	2.93	0.12	278.28	99.86	272	2.93	0.10	320.97	99.89
ZJV3_1	143	0.67	0.77	199.03	99.85	305	2.75	0.18	347.21	99.90
ZJV3_2	123	1.45	0.35	222.04	99.87	316	2.97	0.11	369.81	99.87
ZJV3_3	122	1.71	0.29	260.02	99.86	297	2.91	0.12	331.15	99.91
ZJV4_1	57	0.94	0.58	115.86	99.93	137	2.39	0.24	140.73	99.99
ZJV4_2	67	1.18	0.46	144.22	99.92	163	2.66	0.18	163.70	100.00
ZJV4_3	66	1.28	0.41	139.25	99.93	165	2.55	0.23	165.87	99.99
ZJV5_1	110	1.23	0.40	137.49	99.91	67	0.89	0.72	67.82	100.00
ZJV5_2	307	1.68	0.29	324.33	99.87	88	1.47	0.54	89.53	99.99
ZJV5_3	149	1.40	0.38	166.77	99.91	49	0.80	0.73	51.66	99.99

**Table 3 foods-11-03334-t003:** Dominant OTUs during fermentation with taxonomic annotation against NCBI database.

Band	OTU ID	Closest Relative Species ^a^	Similarity Rate (%) ^b^	GenBank Accession Number
1	OTU5	*Saccharopolyspora rectivirgula*	100.00	JN010261.1
2	OTU31	*Virgibacillus necropolis*	100.00	MT397009.1
3	OTU227	*s__unclassified_g__norank_f__norank_o__Chloroplast*	-	-
4	OTU314	*Acetobacter pasteurianus subsp. pasteurianus*	100.00	MT613452.1
5	OTU325	*Lactobacillus spicheri*	100.00	LC480813.1
6	OTU335	*Lactobacillus acetotolerans*	100.00	CP051649.1
7	OTU336	*uncultured Lactobacillus sp.*	100.00	MN664829.1
8	ZOTU12	*Thermomyces lanuginosus*	100.00	MT316378.1
9	ZOTU21	*Xeromyces bisporus*	100.00	NR_154540.1
10	ZOTU167	*Xeromyces bisporus*	99.60	NR_154540.1
11	ZOTU276	*Alternaria alstroemeriae*	100.00	MT573466.1
12	ZOTU333	*Cladosporium chasmanthicola*	100.00	NR_152307.1
13	ZOTU530	*Aspergillus heterocaryoticus*	100.00	NR_163674.1

^a^ Only the highest homology matches are presented. ^b^ Identity represents % similarity shared with the sequences in the GenBank databases.

## Data Availability

Data are contained within the article or Appendix A.

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
