# Peer review of "Dynamics of Microbial Communities, Flavor, and Physicochemical Properties during Ziziphus jujube Vinegar Fermentation: Correlation between Microorganisms and Metabolites"

_foods, 2022, doi:10.3390/foods11213334_

Round 1

Reviewer 1 Report

The article is well structured, experiments are appropriate, and statistical tools are exploited. However, for this reviewer, some amendments must be addressed. Initially, authors might consider submitting the manuscript for language revision since many grammar and spelling mistakes are present.

Comments:

Line 14: “is” instead of “are”

Line 21: “periods” instead of “period”

Line 37: Grammar mistake

Line 41: Grammar mistake

Line 48: Grammar mistake

Line 72: Please describe accurately what “Daqu” is, how it was obtained, and the added amount in terms of cell concentration.

Line 73: Similar to the previous comment, please specify how bacteria were grown and the added amount in terms of cell concentration.

Line 73: Specify the amount of accessory material added.

Line 76: Grammar mistake

Lines 101-102: Briefly describe the total acid content methodology.         

Line 157-159s: Please support these asseverations with references.

Line 193: Grammar mistake

Line 197: Grammar mistake

The findings of this investigation are well described, and the established correlations are appropriate; however, it would be of interest that the authors include a final remarks section or similar, in which they specify how these findings can be exploited for the quality improvement of ZJV. In brief, it would be interesting to know the practical applications of the investigation findings.  

Reviewer 2 Report

The present study, "Dynamics of microbial communities, flavor, and physicochemical properties during Ziziphus jujube vinegar fermentation: Correlation between microorganisms and metabolites" is well conducted and presented according to the content submitted. This study brings some insights into vinegar production regarding the microbial contributions at the different stages of fermentation and the development of sensory characteristics. The manuscript is fairly prepared but needs some critical improvements. 

There is a need for  language editing 

In your abstract, kindly write OTUs in full as it is the first time it appears.

L25: Remove "etc."

The introduction does not give adequate/sufficient background that would complement the content presented in the discussion. Mainly the background about different vinegar production styles and the evolution of flavor compounds; microbial identification methods and the reason why such an approach was chosen in the current study. 

L57:From those fewer studies, kindly provide information about the microbial dynamics found in those spontaneous fermentation studies. Besides, make it clear that this is spontaneous/inoculated fermentation by actually explaining the concept of spontaneous/inoculated fermentation.

Your methodology lacks information about the microbial origin; inoculum preparation; sample collection during fermentation.

The sensory evaluation in such a study is very critical and you have not given details about your sensory evaluation method and the points used for scoring if any. 

You need to provide information about the chemical composition of the feed used in order to elucidate the fact that indeed the carbon source used was suitable for a vinegar fermentation process. 

Put the part dealing with chemical analysis and product of fermentation analysis before the analysis of microbial diversity, to elucidate the reason behind the observations that are caused by the diversity of the microbial population present. 

L221: Provide the related reference(s)

L229: Avoid using an abbreviation at the beginning of a sentence

In your discussion, you did not compare the current findings to previous similar work where you would then bring the audience to a clear understanding of the novelty of your work. 

A recommendation statement is missing in your Conclusion.

Reviewer 3 Report

In this paper, the Ziziphus jujuba has been used for the fermentation of the vinegar. The authors evaluated the organic substances (organic acids, amino acids) and aroma components (volatile substances) during the Ziziphus jujuba vinegar (ZJV) fermentation. They also evaluated the microbial community during the fermentation, especially Penicillium, Lactobacillus, and Acetobacter. This study is the first paper to explore the relationship between microorganisms, physicochemical indicators and flavor compounds of Ziziphus jujuba.

After a careful reading of the work, I can indicate that the work is well done, precise in every part, but minor revision is needed.

Line 59: change  “had been used” with "has been used"

Line 76: when you say: The acetic acid fermentation was cost 14 d. Samples were collected at the fermentation days of 1, 7, 12, 17 and 22… isn’t clear. You say that you make the fermentation for 14days and that you collect at 17 and 22 days. The times are different

Line 76: change was cost with has cost
